# Effective surveyed area and its role in statistical landslide susceptibility assessments

Txomin Bornaetxea[1], Mauro Rossi[2], Ivan Marchesini[2], and Massimiliano Alvioli[2]

[1]Department of Geography, Prehistory and Archaeology, Faculty of Arts of the University of the Basque Country UPV/EHU. c/ Tomás y Valiente, s/n, 01006, Vitoria-Gasteiz.
[2]Consiglio Nazionale delle Ricerche, Istituto di Ricerca per la Protezione Idrogeologica, via Madonna Alta 126, 06128 Perugia, Italy.

**Correspondence:** Txomin Bornaetxea (txomin.bornaetxea@ehu.eus)

**Abstract.** Geomorphological field mapping is a conventional method to prepare landslide inventories. The approach is typically hampered by the accessibility and visibility, during field campaigns for landslide mapping, of the different portions of the study area. Statistical significance of landslide susceptibility maps can be significantly reduced if the classification algorithm is trained in unsurveyed regions of the study area, for which landslide absence is typically assumed, while ignorance about landslide presence should actually be acknowledged. We compare different landslide susceptibility zonations obtained by training the classification model either in the entire study area or in the only portion of the area that was actually surveyed, which we name effective surveyed area. The latter was delineated by an automatic procedure specifically devised for the purpose, which uses information gathered during surveys, along with landslide locations. The method was tested in Gipuzkoa Province (Basque Country), North of the Iberian Peninsula, where digital thematic maps were available and a landslide survey was performed. We prepared the landslide susceptibility maps and the associated uncertainty within a logistic regression model, using both slope units and regular grid cells as reference mapping unit. Results indicate that the use of effective surveyed area for landslide susceptibility zonation is a valid approach to minimize the limitations stemming from unsurveyed regions at landslide mapping time. Use of slope units as mapping units, instead of grid cells, mitigates the uncertainties introduced by training the automatic classifier within the entire study area. Our method pertains to data preparation and, as such, the relevance of our conclusions is not limited to the logistic regression but are valid for virtually all the existing multivariate landslide susceptibility models.

## 1 Introduction

Landslide susceptibility is defined as the likelihood of a landslide occurring in an area on the basis of the local terrain and environmental conditions (Brabb, 1984; Guzzetti et al., 2005). Landslide Susceptibility Zonation (LSZ) is important for landslide mitigation plans, since it supplies planners and decision makers with essential information (Van Den Eeckhaut et al., 2012). A large number of LSZ studies based on statistical methodologies (Reichenbach et al., 2018) and comparative studies (Cascini, 2008; Das et al., 2010; Schicker, 2010; Amorim, 2012; Blais-Stevens et al., 2012; Trigila et al., 2015; Wang et al., 2015) were published in the last decades. Many statistical methods, aimed at estimating the propensity of a territory to experience

slope failures, rely on landslide inventory maps and spatial thematic layers as predisposing factors (Ermini et al., 2005; Van Den Eeckhaut et al., 2006; Camilo et al., 2017).

In statistical landslide susceptibility models, as the logistic regression (LR) model adopted in this work, the preparation of the training data set is a fundamental and critical step. Commonly, this requires the selection of a sample of stable (without landslides) and unstable (with landslides) mapping units. While assuring the presence of a landslide is straightforward, and it can be supported by the geomorphological signatures on the slope or by direct observation of the events, the selection of landslide-free areas is more critical. Assuming as landslide-free the locations of a study area where no landslides were reported in a field survey is correct only in the unlikely circumstance that the landslide inventory has been prepared surveying every single site of the study area, and following homogeneous criteria. In other words, any landslide-free location in an inventory map should have been explicitly checked to be free from landslides.

Nowadays, there are methods based on the visual interpretation of aerial photographs or digital processing of remotely acquired optical and radar imagery (Catani et al., 2005; Herrera et al., 2009; Fiorucci et al., 2011; Casagli et al., 2017; Mondini, 2017; Fiorucci et al., 2018; Alvioli et al., 2018b), that allow to prepare historical and event landslide inventories. However, the adoption of such methods can be hampered by the lack of imagery or image interpretation expertise, low performance of automatic classification, and other factors. Alternatively, bibliographic sources like newspapers and news feeds, administrative reports or scientific literature can be used to obtain landslide information. Nevertheless, the downside of these type of data is that they hardly are as accurate as required by LSZ studies. As a consequence, sometimes the best option to obtain a reliable landslide inventory is a straightforward geomorphological field mapping. A detailed discussion about the characteristics, advantages and limitations of different approaches for landslide mapping can be found in Guzzetti et al. (2012); Santangelo et al. (2015); Fiorucci et al. (2018).

An operational disadvantage of field-based landslide mapping is the difficulty in surveying the whole area where the LSZ must be carried out since some places can be inaccessible or not visible from the accessible places. Difficulties in surveying the landscape affect the completeness and the spatial representativeness of the landslide inventory and, as a result, inclusion of non-visible areas within a landslide inventory introduces a bias since presence or absence of landslides cannot be ascertained in such portions of landscape. This uncertainty has hardly been considered in existing studies that use field-based landslide inventories (Yesilnacar and Topal, 2005; Murillo-García et al., 2015; Wang et al., 2017).

On the other hand, selection of an appropriate terrain subdivision is also a critical step in LSZ analysis. The land surface can be divided in portions following geomorphologic features using terrain units, topographic units, geo-hydrological units or slope units, but also considering thematic layers resulting in unique condition units or administrative units, as well as regular grid cells partitions (Van Den Eeckhaut et al., 2006; Reichenbach et al., 2018). Selection of different mapping units can result in considerable differences in the susceptibility assessment (Carrara et al., 2008). In this work, we considered grid cells and slope units (Carrara et al., 1991, 1995; Guzzetti et al., 2006; Alvioli et al., 2016; Zêzere et al., 2017; Rosi et al., 2018; Ba et al., 2018), and investigated the effect of the different ways of training LSZ models within both types of mapping units.

We propose an automatic and reproducible procedure to delineate the actual area which was explicitly surveyed in preparing a landslide inventory by geomorphological field mapping, *i.e.* the effective surveyed area (ESA), and to use such relevant

information in statistical analyses. The procedure allows to carry out the calibration of a statistical model within the ESA and then to apply the resulting susceptibility model to the whole area (WA) under investigation. Moreover, we implemented an automatic approach for the delineation of the ESA in a newly developed GRASS GIS module named *r.survey.py*. The software delineates the theoretical visible areas from the points of view recorded during a field campaign by the GPS tracks. Most importantly, the ESA delineated by r.survey.py is an objective and reproducible portion of the study area directly observed by the geomorphologists, thus allowing to avoid arbitrary assumptions about which sites were actually surveyed and which ones were not.

This work aims at demonstrating that the calibration of a landslide susceptibility model within the ESA, instead of the WA (the whole study area, encompassing the ESA), enhances the performance of model itself. In a test study area, we calibrated the multivariate logistic regression model for landslide susceptibility in four different ways, combining two different calibration areas (ESA and WA) with two different mapping unit types: (i) a regular grid cell partition with a ground resolution of 5 m x 5 m and (ii) a slope unit (SU) partition (consisting in irregular terrain subdivisions bounded by drainage and divided lines).

The paper is organized as follows. Section 2 provides an overview of the study area. Section 3 shows the details about data acquisition; in particular, the r.survey is described in 3.3 and SU delineation in 3.4. Section 4 contains a general description about the multivariate method applied to model landslide susceptibility and the approach followed to validate model results, as well as a detailed description about the set-up of the different model assessments. Results are described in Section 5, and are further discussed in Section 6. Eventually, our conclusions are drawn in Section 7.

## 2   Study Area

The Gipuzkoa Province was selected as test study area. It is located in the north of the Iberian Peninsula along the western end of the Pyrenees, and covers an area of 1980 km$^2$, with altitude ranging from the sea level to 1528 m a.s.l. Six watersheds of different size drain the study area and reach the Cantabrian Sea (Fig. 1a). The Province is characterized by a steep morphology with 55% of its surface having a slope larger than 15°.

The investigated area is lithologically heterogeneous (Fig. 1b), with materials ranging from Paleozoic rocks to Quaternary deposits (EVE, 2010), and it corresponds to a hilly and mountainous Atlantic landscape (Mücher et al., 2010). The average annual precipitation is 1,597 mm (González-Hidalgo et al., 2011) with two maximum periods: 34% in November-January and 10% in April. Even though rainfall is the primary triggering factor of shallow landslides (Petley et al., 2005; Alvioli et al., 2018a), anthropogenic slope modifications such as slope clearings and forest extraction activities also strongly affect landslide occurrence (Corominas et al., 2017) in the area.

## 3 Data preparation

### 3.1 Landslide inventory

We prepared a landslide inventory by a direct geomorphological field survey, during the period from June to August, 2016. We collected information about the location of each observed landslide, four GPS points (crown, toe, and two flanks), pho-
5 tographs, features of the surrounding area and information about the landslide type, according to the Cruden and Varnes (1960) classification. Each documented landslide was drawn and digitized using its four GPS waypoints recorded and photographs as a reference. The QGIS software and Google Earth satellite imagery were used for the purpose. Moreover, and most importantly to define the ESA, we digitized the route followed during the field survey. This information was then elaborated using a GRASS GIS module developed for the purpose and included in this work as supplementary material.

As a result of several field trips, 793 individual landslides were collected; 746 of them were classified as shallow movements (Fig. 2a). Our observations together with the existing literature (INGEMISA, 1995; Gipuzkoako Foru Aldundia, 2013; IDE de Euskadi, 2014) confirmed that shallow slides are the most frequent type of landslide in the study area. Consequently, in order to consider only landslides triggered by the same mechanisms, only shallow movements were used to determine landslide presence when defining the dependent variable in the susceptibility assessment. Figures 2b and 2c show the distribution of
landslide sizes, highlighting that a difference of five orders of magnitude exists between the smallest and the largest inventoried shallow slide.

### 3.2 Explanatory Variables

The selection of the appropriate explanatory variables to build a landslide susceptibility model is an important step (Ayalew and Yamagishi, 2005; Schlögel et al., 2018), and no universal criteria nor guidelines exist for the purpose.
We obtained relevant environmental digital layers from the Spatial Data Service of the Basque Country[1], and created 13 maps describing the different explanatory variables (see Table 2). To produce derived morphometric continuous variables, such as *Slope*, *Sinusoidal slope*, *Surface area ratio (SAR)*, *Terrain wetness index (TWI)*, *Curvature*, *Plan curvature* and *Profile curvature*, we used a DEM raster layer with 5 m x 5 m spatial resolution. *Sinusoidal slope* is a derived morphometric variable proposed by Santacana Quintas (2001) and Amorim (2012) to emphasize the fact that shallow slides typically occur in medium
slope areas, while they seldom occur on slopes steeper 45°. For categorical variables, such as *Lithology*, *Permeability*, *Regolith thickness*, *Land Use*, *Vegetation and Aspect*, we computed frequency ratio (FR) values for each class, and used them as a relative value for their transformation into continuous variables (Lee and Min, 2001; Yilmaz, 2009; Trigila et al., 2015). We acknowledge that the FR values can vary depending on the portion of the territory considered as the total area (ESA or WA). In order to perform a direct comparison, we decided to maintain the same FR values (calculated considering the WA) in all
regular grid cell-based susceptibility analysis.

---

[1]http://www.geo.euskadi.eus

In this work, we first adopted grid cells as mapping units, and we applied a simplified and statistically oriented work-flow that ensured that only significant variables were taken into account as well as the non-redundancy of the contributed information by each covariate (Ayalew and Yamagishi, 2005). To do so, the whole set of 13 variables were considered within the LR analysis, and correlation coefficients were computed. We considered as collinear two variables when their correlation coefficient is greater than 0.5 with a significance level of 0.01. In such a case, as an objective criterion for variable selection, the variable with highest p-value between the two (see section 4.1), was not taken into consideration in the final run of the susceptibility LR model. Additionally, variables with p-value higher than the threshold of 0.05 were rejected.

Then, considering the variables actually used for the application of the statistical models with grid cells, we have further restricted the set of variables to be used with slope units (see section 5.2).

## 3.3 Definition of the effective surveyed area

In this work we suggest the concept of ESA, and training of statistical models therein, as an approach to be used to train a landslide susceptibility model avoiding assumptions about the presence or absence of landslides in areas not explicitly observed. We delineated the ESA by means of the newly developed GRASS GIS python module *r.survey.py* (see supplementary material). Input data to define the visible area (*i.e.* ESA in our case) are: i) a sample of points to be considered as points of view; ii) a DEM of the area; iii) the maximum visible distance. The sample of points of view, in our case, was defined re-sampling a given number of points along the recorded path during the field campaigns. This number of points depends on the maximum distance set between them, and together with the DEM resolution selected the results can be directly affected. In a 10 km$^2$ subset of the study area, we tested the software output using: i) maximum distance between sampled points of 50, 100, 200 and 500 m; ii) the original DEM at 5 m resolution and resampled versions of the DEM at 20, 50 and 100 m resolution; iii) maximum visible distance of 500 m (the later was dictated by the largest distance between the digitized field path and the farthest landslide pixel in the subset of the study area). Results of the test are summarized in Table 1.

We considered that the best setting option was the one which allows covering the totality of the landslides using the smallest number of points (larger $D_{max}$ value) and the lower DEM resolution in order to optimize the calculation time. In our case, considering the whole study area, the maximum visible distance was set to 1,100 m, in view that the largest distance between the digitized field path and the farthest landslide pixel was 1,092 m. Then, and according to the results of Table 1, we set the maximum sampling distance to 200 m and adopted a DEM resolution of 100 m.

We can make sense of the numerical values of the parameters used in the *r.survey.py* module considering that the minimum size $A$ of an object visible from a distance $\Delta$ is given by Rodrigues et al. (2010) and Minelli et al. (2014):

$$A = \frac{25 \, \Delta^2}{c}, \tag{1}$$

where $c$ is a steradiant to square minutes conversion factor, $c \simeq 1.18 \cdot 10^7$. Using $\Delta = 1,100$ m in Eq. (1), we get $A = 2.6$ m$^2$, meaning that the smallest landslide in our inventory, with size 7.3 m$^2$, would actually be identifiable from at least one point along the route, if the landslide sits within the ESA. The resulting ESA covers 44.24% of the entire study area and it is shown in Fig. 2a.

## 3.4 Slope units delineation

For SU delineation we have adopted the *r.slopeunits* software described in Alvioli et al. (2016). The software is a GRASS GIS module, as the *r.survey.py* code presented in this work, and it was designed for the automatic and adaptive delineation of SUs, given a DEM and a set of user-defined input parameters. The code can be used to produce several SU partitions, using different combinations of the input parameters, which can thus be tuned according to user-defined criteria. We partially followed Alvioli et al. (2016), in that we selected the best SU partition considering the quality of terrain aspect segmentation. In addition, we have performed preliminary tests using the LR susceptibility model, showing that the use of very small SUs provides unrealistic results, which can be understood considering the limited variability of variables within such small SU polygons. We concluded that, in the case of the Gipuzkoa Province the most suitable SU partition for landslide susceptibility zonation should be obtained with the following *r.slopeunits* input parameters: flow accumulation area threshold $t = 1$ km$^2$; minimum SU planimetric area $a = 0.15$ km$^2$; minimum circular variance of terrain aspect within each SU $c = 0.2$; reduction factor $r = 5$; threshold value for the cleaning procedure *cleansize* = 0.025 km$^2$. As a result, we obtained a set of SUs which range in size from 0.026 km$^2$ to 3.6 km$^2$ with average 0.28 km$^2$. A discussion of SU delineation and optimization of input parameters can be found in Alvioli et al. (2016) and Schlögel et al. (2018), and it is out of the scope of this work.

## 4 Modelling framework

We prepared four landslide susceptibility maps (LS maps), by means of a multivariate LR model. Classification performances were measured by means of a set of validation tests explained in the following sections. We prepared the first two maps using 5 m x 5 m regular grid cells as mapping units. The two maps differ because in one case the LR model was calibrated within the WA, and within the ESA (described in Section 3.3) in the other case. The third and fourth LS maps, instead, were prepared with different mapping units, namely with SUs (described in Section 3.4) instead of grid cells, where calibration data were also changed considering data within WA in one case and within ESA in the other. We end up with four maps, which we name as follows: WA-PM (whole area, pixel map), ESA-PM (effective surveyed area, pixel map), WA-SUM (whole area, slope units map) and ESA-SUM (effective surveyed area, slope units map).

### 4.1 Logistic regression

We used logistic regression (Hosmer Jr et al., 2013), one of the multivariate statistical approaches available in the LAND-SE software (Rossi and Reichenbach, 2016), to build the landslide susceptibility model in the test study area. The method is the most used in the scientific literature (Reichenbach et al., 2018) and proved to be useful and reliable in several studies (Nefeslioglu et al., 2008; Van Den Eeckhaut et al., 2012; Trigila et al., 2015). The LR model works with either continuous or categorical independent variables, or a combination of the two types, regardless whether they are normally distributed or not (Costanzo et al., 2014).

The mathematical relationship between the dependent dichotomous variable (presence/absence of a landslide in the mapping unit; $Y$ in the following) and the $n$ independent variables (e.g., slope, lithology, etc.; $X_1, ..., X_n$), within the LR model, reads as follows:

$$Y = \beta_0 + \beta_1 X_1 + ... + \beta_n X_n,  \tag{2}$$

where $\beta_0$ is the intercept of the model and $\beta_1, ..., \beta_n$ the linear regression estimate coefficients. The independent (explanatory) variables, $X_1, ..., X_n$, included in our case both continuous and categorical layers (the latter were previously transformed into continuous variables, as described in Section 3.2); see Table 2 for the full list of variables used in this work. Calibrating an LR model amounts to selecting numerical values for the $\{\beta_i\}_{i=1}^{i=n}$ coefficients in Eq. (2) that maximize the agreement between model output, i.e. landslide probability:

$$P = \frac{1}{1 + e^{-Y}},  \tag{3}$$

and empirical landslide data, in a training area. The same values of the coefficients can then be used to validate the model prediction skills in a different area, where landslide conditions are unknown to the model but the same explanatory variables layers exist.

In addition to the $\beta$ coefficients, the LR method also offers a significance p-value for each explanatory variable. The implementation of the `glm` function of the R programming language library[2], used in the LAND-SE software, is such that it is possible to investigate the estimated standard error of a t-statistic for the null hypothesis of each of the coefficients of the linear model. The p-value represents the probability for the parameter to be zero: for p-values smaller than 0.05 the null hypothesis (vanishing coefficient) is rejected, thus the associated variable is significant for the final result. So, the p-value can be considered as an objective indicator for the selection of the most relevant variables to be used in the statistical model (Schlögel et al., 2018).

## 4.2 Evaluation of model performance

The performance of statistical susceptibility models, *i.e.* of multivariate binary classifiers, can be evaluated comparing their predictions with the landslide data used in the model calibration/training step (*i.e.* model fitting performance) or with independent landslide data (*i.e.* model prediction performance). The definition of training and validation input samples is crucial to detect how well each model fits input data, but also how good is the model at predicting new data.

The statistical metrics commonly used in the literature (Corominas and Mavrouli, 2011; Van Den Eeckhaut et al., 2006; Lombardo et al., 2015; Reichenbach et al., 2018) for that purpose are (i) confusion matrices (contingency tables) and their graphical representation (four-fold or contingency plots), (ii) Receiver Operating Characteristic (ROC) curves and their associated Area Under Curve ($AUC$) value, (iii) classification error plots and (iv) Cohen's kappa index.

Four-fold (or contingency) plots are visual representations of the confusion matrices reporting the percentages of the true positives ($TP$), true negatives ($TN$), false positives ($FP$) and false negatives ($FN$). ROC curves are more complex represen-

---

[2]https://www.r-project.org/

tation of the classification performance based on different probabilistic threshold values. The area under the ROC curve ($AUC$) is an indicator of the model performance in predicting landslide susceptibility. $AUC$ values vary between 0 and 1, with higher values indicating better prediction skills (Fawcett, 2006).

To estimate the uncertainty associated with the landslide susceptibility value assigned to each mapping unit, it is possible to run multiple instances of the LR model varying, randomly, the input data. In each run, the input is prepared sampling the original training data set with a bootstrap technique, consisting in a random sampling with replacement (Efron, 1992; Davison and Hinkley, 1997; Rossi et al., 2010; Rossi and Reichenbach, 2016). Classification error plots summarize the distribution of multiple results and show the mean probability estimate of landslide spatial occurrence for each mapping unit (x-axis), ranked from low (left) to high (right) values, related to the variation of the model estimate (y-axis), measured by 2 standard deviations ($2\sigma$) of the probability estimates obtained by the different model runs (Guzzetti et al., 2006). The parabolic model fitting equation resulting from the point cloud (*i.e.* using a non-linear least square method), describes analytically the overall model prediction performance variability. Cohen's kappa index ($k$) is an additional measure of the reliability of a classification model (Cohen, 1960; Rossi et al., 2010), whose higher values also indicate a more accurate prediction skill.

In this study the probability of landslide occurrence resulting from each model estimate (trained either within the ESA or within the WA) and for each considered mapping unit (either grid cells or slope units), was reclassified in five landslide susceptibility classes which were labelled as Very low (for susceptibility values in the range 0-0.2), Low (0.2-0.45), Medium (0.45-0.55), High (0.55-0.8) and Very high (0.8-1).

Moreover, in order to spatially identify the pairwise matching degree between different model estimates, we additionally adopted a simplified classification of the landslide susceptibility. Each mapping unit was reclassified as stable or unstable considering a threshold value of 0.5. The different maps, all prepared with the same mapping unit partition, were overlapped. Then, the mismatch degree between grid cell and SU susceptibility maps was quantified in terms of number of mismatched mapping units and overall mismatched area.

## 4.3   Data selection for landslide susceptibility

The DEM available for the study area consists of $7.91 \cdot 10^7$ cells with 5 m resolution. For landslide susceptibility assessment, both using grid cells (*i.e.* pixels) and SUs, we prepared raster layers corresponding to each available explanatory variable, aligned to the DEM grid cells.

We devised a rigorous sampling procedure to minimize possible statistical biases during training/validation partition. The procedure is slightly different for the grid cell and SU mapping units cases.

In the first case, a grid cell was considered unstable if it is located within any landslide polygon, and stable if it is outside the landslide boundaries. In the second case, an SU was considered unstable depending on the percentage of landslide area present within it. In any case, the 75% of the unstable mapping units together with a similar amount of stable mapping units were used to train the LR model, and the remaining 25%, also together with a similar amount of stable mapping units, for validation. The choice of an equal number of stable and unstable mapping units was done on purpose, and it is the standard procedure required

by the LAND-SE software for landslide susceptibility assessment, because the LR model requires a balanced data set, in which the number of stable and unstable cases are similar (Felicísimo et al., 2013; Costanzo et al., 2014).

For regular grid cell-based models, we selected at random 558 landslides (75%) for model training, and converted them into raster layers (84,623 unstable pixels). The remaining 188 landslides (25%), used for validation, were also rasterized (29,247 unstable pixels). This is at variance with the usual random selection of unstable pixels, in which a given percentage of grid cells are sampled within landslide polygons. Here we select whole landslides, and consider all the pixels encompassed by the landslide bodies as training/validation samples. We ran the experiment with three different training/validation random sets, containing the above percentages. This exercise allowed us to confirm that the random selection of the landslide inventory does not affect the model results in a relevant way, because in all the cases the model classification performances were very similar. In order to choose one single data set for further comparative analyses, the data set with the best classification result was selected. Then, training sets were selected as follow: 84,623 unstable pixels and an equal number of stable pixels were selected as training sets. Two different sets were selected at random first within WA and then within ESA. We made sure that unstable pixels were exactly the same in the two cases, because we wanted the only difference to be that the stable pixels were sampled within the WA, in the first case, and within the ESA, in the second case. Finally, in order to guarantee the comparability of the prediction performances, one unique validation sample was created as follow: the remaining 29,247 unstable pixels together with an equal number of stable pixels selected at random within the remaining stable pixels within the ESA.

Concerning the SU-based models, we first partitioned the study area in 6,907 SUs with the technique outlined in Section 3.4. SU boundaries do not match those of the dependent or explanatory variables layers, allowing the presence of different classes, or values, inside each SU. Moreover, the presence of one single landslide pixel within a slope unit was not considered enough to label this SU as unstable. Therefore, instead of arbitrarily defining a given threshold value in order to consider an SU as unstable, we decided to use the overall landslide density in the WA. For this reason, we considered as unstable those SUs containing 0.15% or more unstable pixels, and stable otherwise. We used as explanatory variables the mean and the standard deviation of the morphometric variables for each SU and the percentage of the area covered by each class of the categorical layers. In 304 cases the SU contained 0.15% or more unstable pixels, so we selected at random 228 of them (75%) for training, and the remaining 76 (25%) were used for validation. Like in grid cell approaches, we created two different training samples where unstable SUs were exactly the same, and only the stable SUs vary in each case. The first training sample includes 228 stable SUs selected at random along the WA. The second training sample includes an equal number of stable SUs units selected at random among those that at least partially overlap the ESA. Additionally, 76 SUs labelled as unstable were selected from the whole set, for validation. The validation sample was completed by adding a random selection of the same number of SUs labelled as stable and which at least partially overlap the ESA. Thus, the validation sample contained 152 SUs (76 unstable + 76 stable).

Eventually, since the ESA is an approximation of the real surveyed area, we stress that we always selected stable mapping units for validation only if they are fully or partially within the ESA, because no evidence exists that a mapping unit falling entirely outside the ESA is actually free from landslide. Moreover, if a portion of an SU falls within the ESA, that implies

that at least one part of the SU was observed. Therefore, using this approach, we can remove at least those SUs that were not surveyed at all.

## 5 Results

### 5.1 Susceptibility maps using grid cells

5  We ran the LR model using the pixel-based data sets twice: once using the entire training pixel sample and once using the effective training pixel sample as dependent variables. We defined the obtained results as whole area pixel map (WA-PM) and effective surveyed area pixel map (ESA-PM), respectively.

Both in WA-PM and ESA-PM, we first used the same 13 explanatory variables, listed in Table 2, and then we selected for each model assessment the most relevant explanatory variables considering the collinearity between each pair of variables and 10  the significance (p-value) of the regression estimates (see section 3.2). As a result, for each case, only the variables marked with an asterisk in Table 2 were introduced in the final LR analysis.

Using the validation pixel sample, we evaluated the prediction skills of the pixel susceptibility maps. Inspection of the four fold, or contingency, plots (Figs. 3a, d) reveals that WA-PM predicted correctly the 63.58% (TP+TN) of the observed unstable and stable mapping units, whereas ESA-PM was capable to correctly predict a higher amount of mapping units (65.45%). The 15  ROC curves (Figs. 3b, e) also indicate better prediction skills in ESA-PM ($AUC = 0.7$) than in WA-PM ($AUC = 0.68$) and the same happens for the Cohen's Kappa index (Fig. 3; $k = 0.309$ versus $k = 0.272$). Moreover, the classification error plots (Figs. 3c, f) provide an estimate of the error associated with the predicted susceptibility values, which does not exceed 0.1 standard deviations in any case, highlighting the reliability of the results. And finally, the mutual mismatch map (Fig. 5e) shows that 14.8% (corresponding to an extension of 293 km$^2$) of the mapping units flipped their landslide susceptibility class in WA-PM 20  and ESA-PM.

### 5.2 Susceptibility maps using slope units

Due to the subdivision of categorical variables in classes, and to the use of mean and standard deviation of morphometric variables, the introduction of the original 13 explanatory variables would result in 56 new variables in which many of them (all those classes belonging to the same categorical variable) would be highly correlated. For this reason, the variable selection 25  approach used in the pixel-based case is not viable when working with SUs and a specific variable selection approach for SU models would require further investigation. Thus, for this work, the most appropriate set of explanatory variables, among those considered as the most relevant in pixel-based model assessment, was selected by expert criteria. Considering such set of variables as a starting point, we selected new sets of explanatory variables to evaluate landslide susceptibility using SUs, *i.e.* to calculate the whole area slope unit map (WA-SUM) and the effective area slope unit map (ESA-SUM). Taking into 30  account that the automatic procedure for the SUs definition already included the flow accumulation calculation, used for *TWI* estimation, and the aspect component, we rejected *Aspect* and *TWI* to avoid spurious correlations. We selected the following

set of variables used to produce both pixel-based maps such as *Lithology*, *Permeability*, *Regolith thickness* and *Vegetation*, and we added *Slope*. The reason for choosing *Slope* over *Sinusoidal slope* or *SAR* is due to the fact that these two are derivative variables of the former. Moreover, we consider *Slope* more suitable feature to describe the average morphology within SU than *Sinusoidal* slope or *SAR*, so we decided to select it in order to simplify interpretation of the results.

Using the validation SU sample, we assessed the prediction skills of the SU maps. For the WA-SUM the 65.13% of the 152 validation mapping units were correctly classified (TP+TN) (Fig. 4a). The ROC curve provides $AUC = 0.69$, and the corresponding Cohen's Kappa is $k = 0.302$ (Fig. 4b). Concerning the classification error plot (Fig. 4c), it can be observed that in the SUs with High and Low landslide susceptibility probability (probability $> 0.8$ and $< 0.2$) the $2\sigma$ value stays below 0.2, but variability in the estimates becomes larger for intermediate susceptibilities. This reveals a considerable variation in the stable/unstable classification of the territory, which implies a low reliability, at least for the intermediate probabilities (Guzzetti et al., 2006). For the ESA-SUM, 63.82% of the 152 validation mapping units were correctly classified (TP+TN) (Fig. 4d) with $AUC = 0.71$, slightly larger with respect to the other SU model assessment, whereas, the Cohen's Kappa index performed slightly worse, being $k = 0.276$ (Fig. 4). The classification error plot shows a considerable variation in intermediate probabilities (Fig. 4f) while the uncertainty is lower for High and Low probabilities. Nevertheless, the quadratic fit curves indicate a lower overall variability for ESA-SUM than for WA-SUM.

Visual inspection of the SU susceptibility maps (Figs. 5b, d) shows similarities between WA-SUM and ESA-SUM. The difference is graphically presented through the mismatch map (Fig. 5f), where 12.6% of the mapping units (corresponding to an extension of 247 km$^2$) change their landslide susceptibility class, between WA-SUM and ESA-SUM.

## 6   Discussion

The number of scientific publications focusing on landslide susceptibility zonation has notably increased during the last decades (Gutiérrez et al., 2010; Rossi and Reichenbach, 2016; Liberatoscioli et al., 2017; Valagussa et al., 2017; Zhou et al., 2018; Reichenbach et al., 2018) and, nowadays, there is a huge variety of applications and comparisons which provides an enormous range of approaches to prepare a landslide susceptibility map. Differences between these approaches can be summarized in (i) the type of landslide inventory, (ii) the environmental variables used, (iii) the mapping unit partition, (iv) the method used to prepare susceptibility maps and (v) the scale of application. The existence of such a big production of papers investigating these aspects is proof that no fully consolidated standard exists for all the steps involved in landslide susceptibility analysis.

In this work, we showed that the information contained in a field-based landslide inventory for landslide susceptibility analysis should be critically examined, also in combination with the mapping unit of choice.

A field work-based landslide inventory is by definition a source of uncertainty in statistical analysis, owing to various reasons, including mapping errors, accuracy, subjectivity, and others. The focus of this work is the analysis of an additional uncertainty due to use of field mapping, namely the fact that it is impossible to ensure that the study area was surveyed in a homogeneous way. An objective delimitation of the surveyed area by means of the ESA, proposed in this paper along with a module to objectively delineate the ESA (see supplementary material), is one way to reduce this uncertainty.

The hypothesis tested in this work is that any statistical landslide susceptibility model trained inside the ESA is by definition more correct than considering the entire study area for training the model. The statement was borne out by the results of multivariate LR model calculations. We acknowledge that the ESA is only an approximation of the real surveyed area, though a much more realistic one than using the whole study area. Our definition of the ESA depends on the maximum distance between points along the field, trips paths and the selected resolution of the DEM. Preliminary tests in a reduced portion of the territory provided the most suitable settings for a satisfactory definition of the ESA in the particular case of Gipuzkoa Province (section 3.3).

In the case of the pixel-based susceptibility maps, the metrics of model prediction performances are in agreement with our main statement about the relevance of ESA. As a matter of fact, all the validation performance tests (confusion matrix metrics, the area under the ROC curve and Cohen's Kappa index) present an improvement if the stable pixels used for training the LR model are selected within the ESA (like in ESA-PM, Fig. 3a) than if they are taken from the WA (like in WA-PM, Fig. 3b). In addition, the almost flat classification error plots in both cases (Figs. 3c, f) show high stability of model results. The spatial distribution of the susceptibility classes are different as well between ESA-PM and WA-PM (see Figs. 5a, c), and such differences are highlighted in the mutual mismatch map (Fig. 5e). Another difference between the two pixel maps is the set of explanatory variables selected as predictors. The variable selection approach presented in this paper, and previously adopted in a similar way in Schlögel et al. (2018), demonstrated to be effective and capable to detect presence of redundant information, as well as offering an objective way to choose between collinear explanatory variables.

In the case of SU-based susceptibility maps, validation metrics do not present us with clear-cut results as in the pixel-based maps. As a matter of facts, $AUC$ performs better in ESA-SUM while Confusion Matrix and Cohen's Kappa index present higher prediction performance in WA-SUM (Fig. 4). The classification error plots show considerable variations in intermediate susceptibility probability values, but the quadratic fit curves suggest a slightly lower variability in ESA-SUM (Figs. 4c, f). We interpret these results as an indication of a smaller effect that proper usage of the ESA can have in SU-based susceptibility maps, with respect to pixel-based maps. Despite the small difference in model prediction performance between WA-SUM and ESA-SUM, the reduction of the mismatch degree (Fig. 5f) suggests that the usage of the ESA is equally recommendable for SU susceptibility maps carried out by field work landslide inventories.

The pixel- and SU-based maps obtained within the method presented in this work are inherently different from a conceptual point of view. We maintain that an SU-based map probably represents a better option, for SUs bear a clear relation with topography, they reduce mapping errors and are more useful for practical (planning) purposes. Nevertheless, for the sake of completeness and to show differences between the two approaches, we discussed pixel-based and SU-based maps independently. The uncertainty introduced by a field work-based landslide inventory can be mitigated by using SUs, resulting in more similar susceptibility maps and validation performances in WA-SUM and ESA-SUM than in pixel models.

Moreover, since the threshold value for distinguishing stable and unstable SUs could affect the LR model performances, we performed a sensitivity test evaluating the LR models, for both the WA and ESA, using different presence/absence thresholds. We carried out calculations using as a threshold the 5th percentile (P$_5$, threshold 0.013%), the 50th percentile (P$_{50}$, threshold 0.265%) and the 90th percentile (P$_{90}$, threshold 4.5%) of areal landslide distribution, along with the average landslide den-

sity calculated within the ESA, i.e. 0.33%. We observed that for all the cases, except in $P_{90}$, the model tests showed better performance for ESA-SUM than for WA-SUM, which is proof that the conclusions obtained following any approach were indistinguishable. We note that because of the high threshold defined in $P_{90}$, the model was trained with a very small sample of unstable SUs, which gives to the result a very poor reliability. On the other hand, for in the $P_5$ case, the unbalance does not take place, since each SU where at least one landslide pixel exists belongs to the unstable class, resulting in minimum yet relevant number of unstable SUs. Therefore, we maintain that results of the test confirm that SUs mitigate the relevance of the calibration area (ESA versus WA) when building an SU-based susceptibility model with a field-based landslide inventory, independently of the landslide presence threshold value. However, we acknowledge that the search of an optimal threshold value that ensures a balanced sample is a relevant point, though it is beyond the scope of this work.

## 7 Conclusions

We explored the effects of training an LR classifier (Rossi and Reichenbach, 2016) for landslide susceptibility zonation within the area that was actually surveyed at landslide mapping time, the ESA, and the extended study area, WA, encompassing the ESA. We prepared four susceptibility maps combining variables (*cf.* Eq. (2) and Table 2) sampled strictly within the ESA or from the WA with two different mapping unit partitions, *i.e.*, 5 m x 5 m grid cells and slope units (Alvioli et al., 2016), delineated for the purpose.

A straightforward comparative analysis using standard prediction performance metrics revealed that the ESA-based approach is better than the WA-based, at least in grid-cell mapping units based approaches, for what concerned the training area (*i.e.* within the ESA or WA). Introducing different mapping units in the comparison, we further found that using slope units slightly reduces the gap between results obtained training a statistical model within the ESA versus WA. Thus, the capacity of the slope unit mapping subdivision to mitigate this error was demonstrated, as suitable alternative to the conventional pixel-based approaches.

The results illustrated above support the following statements:

(i) working with pixel mapping units, training a statistical classifier for LSZ within the ESA is the correct approach to reduce the uncertainty inherent to the landslide inventory;

(ii) working with slope unit terrain partition this uncertainty can be mitigated, even though it is still advantageous to train the LS model within the ESA;

(iii) use of ESA should be considered, if sufficient information is available, in preparing landslide susceptibility maps with any multivariate statistical model;

(iv) collecting information about the path followed during field campaigns for landslide mapping is a meaningful procedure for estimating the ESA, at model assessment time, using the GRASS GIS module *r.survey.py* presented in this work.

We acknowledge that the overall performances of the landslide susceptibility maps presented in this paper are of moderate to low prediction capacity, with $AUC$ values ranging between 0.68 to 0.71 and an overall accuracy which hardly overcomes 65%

in the best case (Figs. 3 and 4). This could be due to (i) the lack of a more complete landslide inventory (Guzzetti et al., 2012; Malamud et al., 2004) or (ii) the use of not up-to-date thematic layers. Nevertheless, the preparation of a definitive landslide susceptibility map for the study area was out of the scope of our investigation. Instead, we performed pairwise comparative analyses in which we only changed, across the compared model assessments, the region of logistic regression training.

*Code availability.*

– The software developed in this work to delineate the effective surveyed area, *r.survey.py*, is contained in the supplementary material

– The software developed in Alvioli et al. (2016) to parametrically delineate slope units, *r.slopeunits*, is available at:
  `http://geomorphology.irpi.cnr.it/tools/slope-units`

– The software developed in Rossi and Reichenbach (2016) for the statistical assessment of landslide susceptibility zonation, LAND-SE,
is available at: `https://github.com/maurorossi/LAND-SE`

*Competing interests.* No competing interests are present.

*Acknowledgements.* This work has been funded by the Cultural Landscape and Heritage UNESCO Chair of the University of the Basque Country (UPV/EHU), and the article could not be possible without the support of the IT1029-16 research group funded by the Basque Government, whose head Iñaki Antigüedad has provided his direct collaboration. Authors are also grateful to professor Orbange Ormaetxea
for her collaboration during the manuscript corrections. Part of this work was carried out by TB during a scientific visit at CNR IRPI.

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

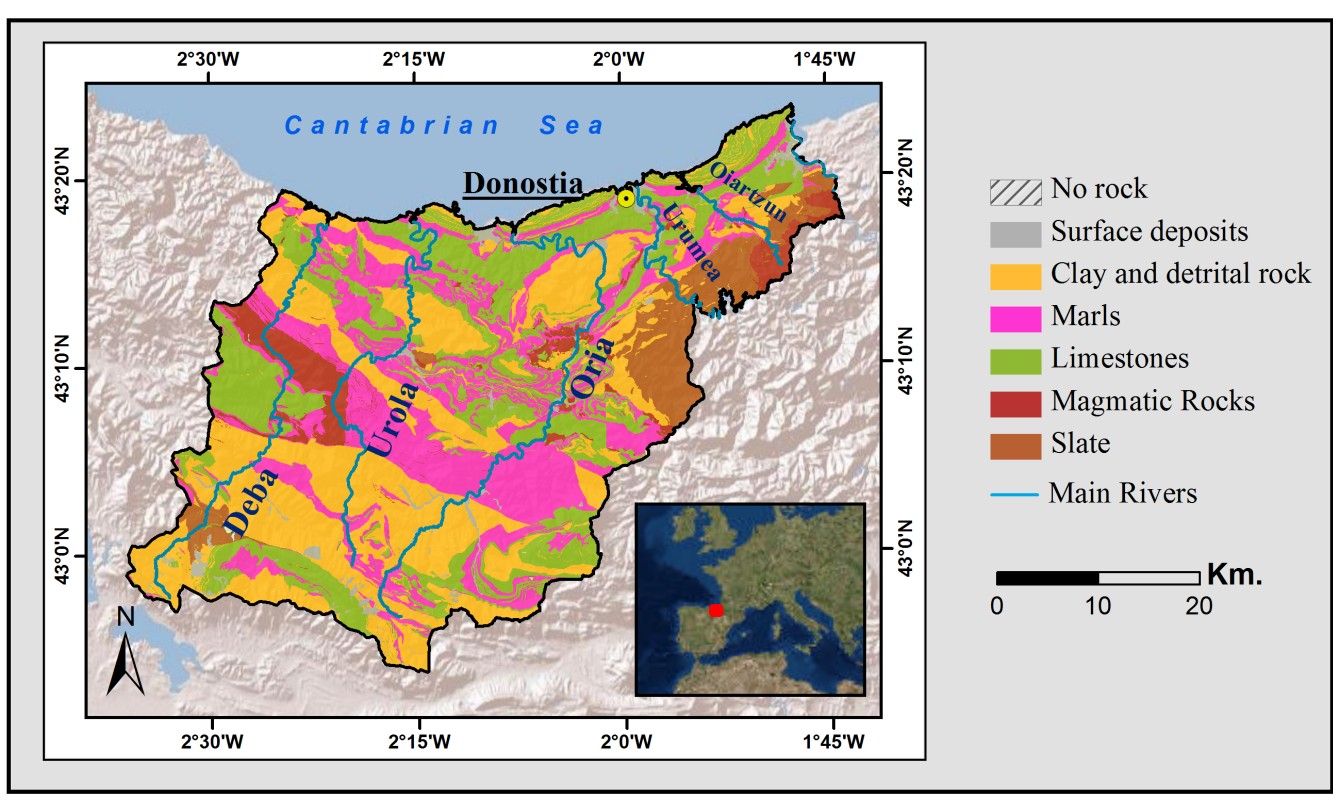

**Figure 1.** Location of the Gipuzkoa Province study area and simplified lithological map developed according to the original map of the Spatial Data Service of the Basque Country. Coordinates in degrees, Universal Transversal Mercator (UTM) Zone 30N, European Datum ETRS 1989.

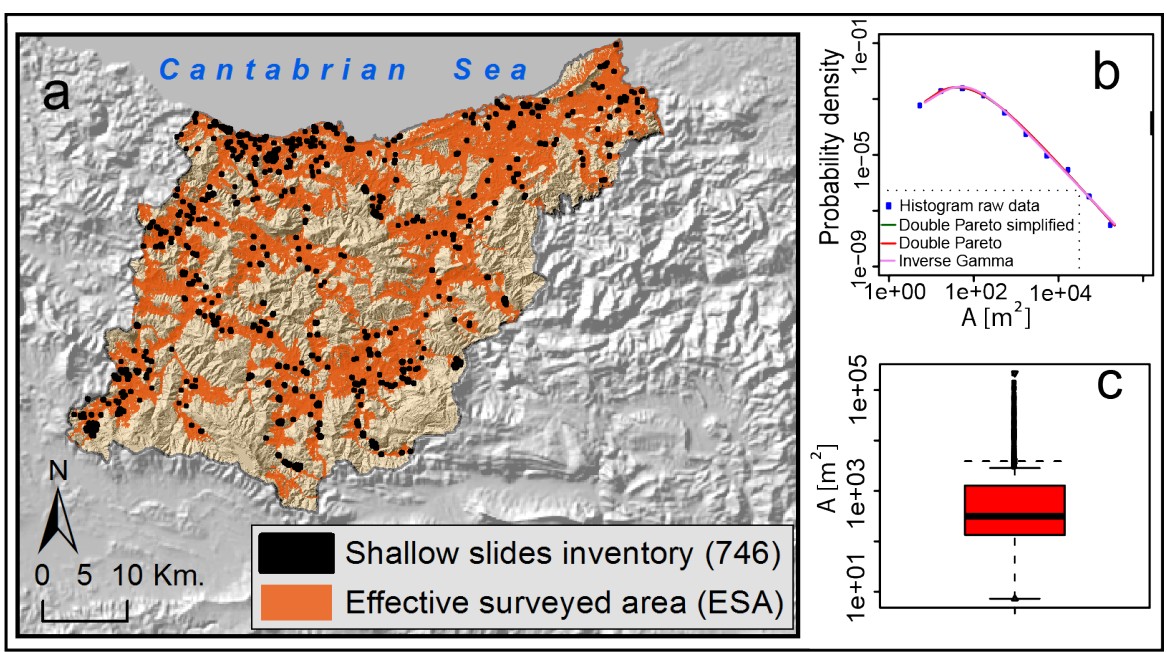

**Figure 2.** (a) Distribution of the shallow slides inventory along the study area and extension of the effective surveyed area (ESA). (b) Probability density plot of the shallow landslide size (Area in m[2]) distribution. (c) Box plot of the same distribution.

## Whole Area Pixel Map (WA-PM)

| Cohen's $k$ | AUC$_{ROC}$ | Overall Accuracy | Overall Error Rate |
|---|---|---|---|
| 0.272 | 0.68 | 63.58% | 36.42% |

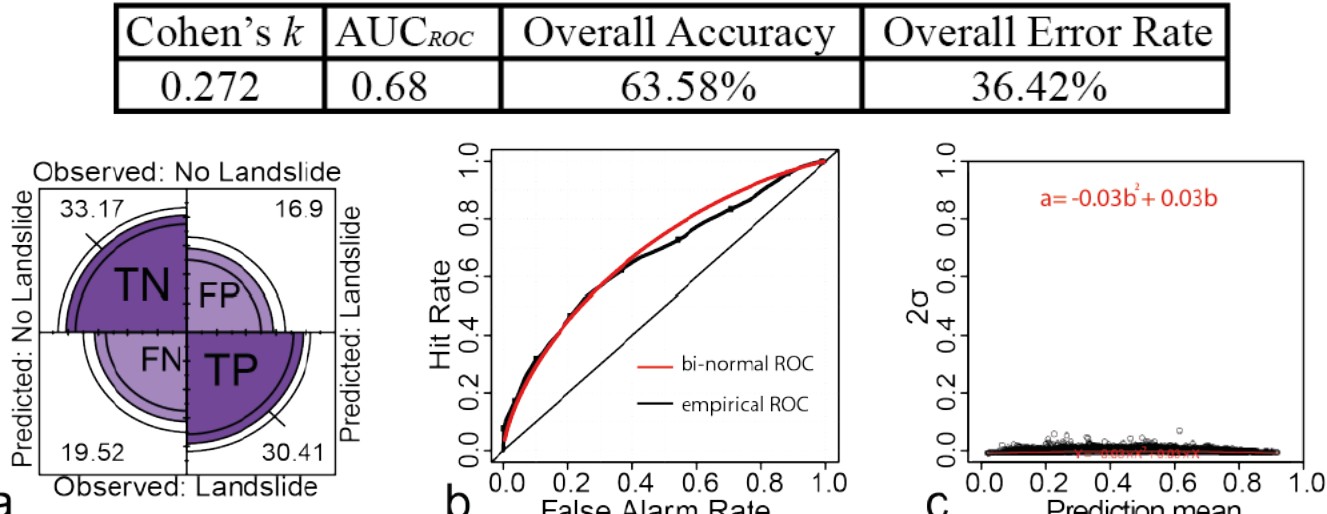

## Effective Surveyed Area Pixel Map (ESA-PM)

| Cohen's $k$ | AUC$_{ROC}$ | Overall Accuracy | Overall Error Rate |
|---|---|---|---|
| 0.309 | 0.7 | 65.45% | 34.55% |

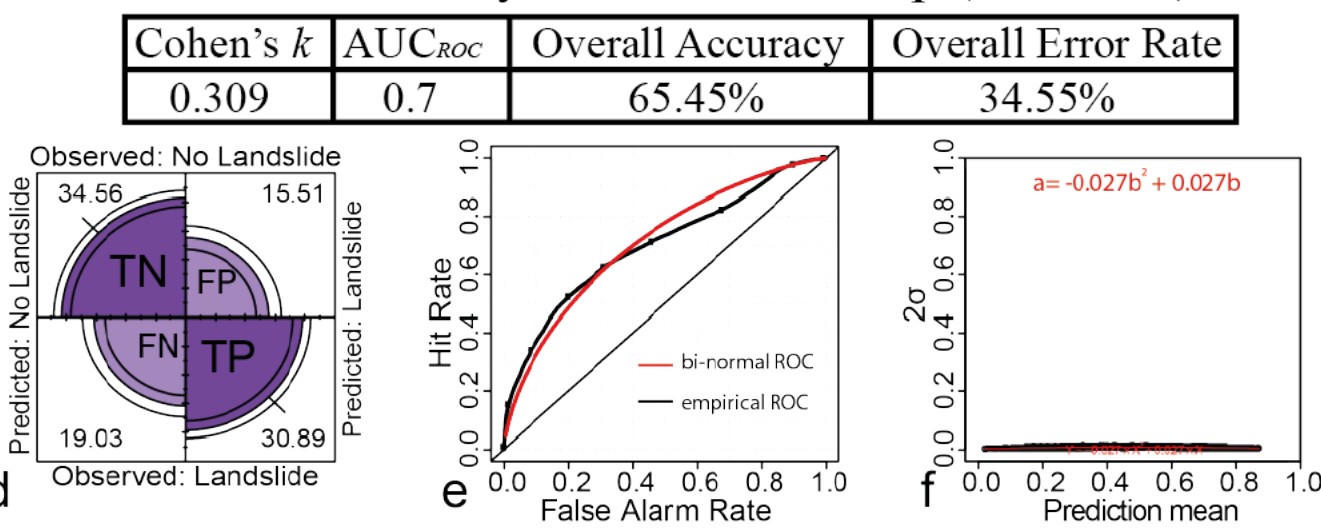

**Figure 3.** Pixel-based LR models prediction performance results: summary tables of the Cohen's kappa index, area under the ROC curve (AUC), overall accuracy ((TP+TN)/(TP+TN+FP+FN)) and overall error rate ((FP+FN)/(TP+TN+FP+FN)); (a,d) four fold or contingency plots; (b,e) ROC curves; (c,f) classification error plots and the quadratic regression fit curves (red line).

# Whole Area Slope Unit Map (WA-SUM)

| Cohen's $k$ | AUC$_{ROC}$ | Overall Accuracy | Overall Error Rate |
|---|---|---|---|
| 0.302 | 0.69 | 65.13% | 34.87% |

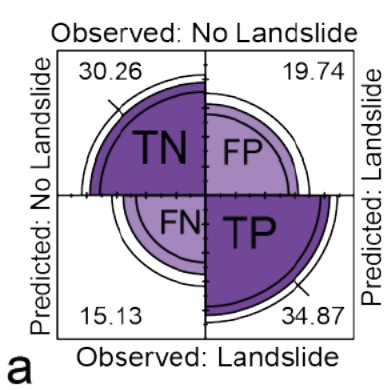

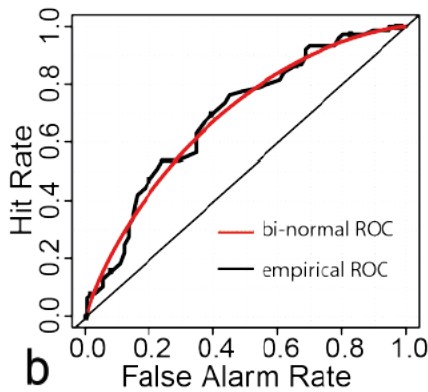

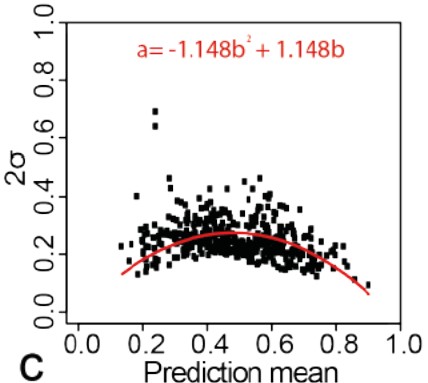

# Effective Surveyed Area Slope Unit Map (ESA-SUM)

| Cohen's $k$ | AUC$_{ROC}$ | Overall Accuracy | Overall Error Rate |
|---|---|---|---|
| 0.276 | 0.71 | 63.82% | 36.18% |

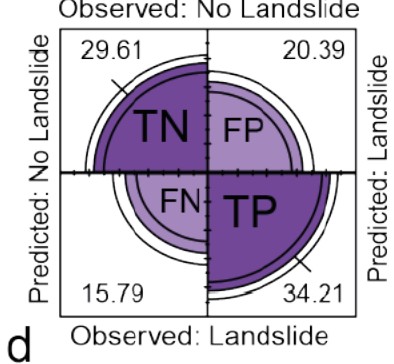

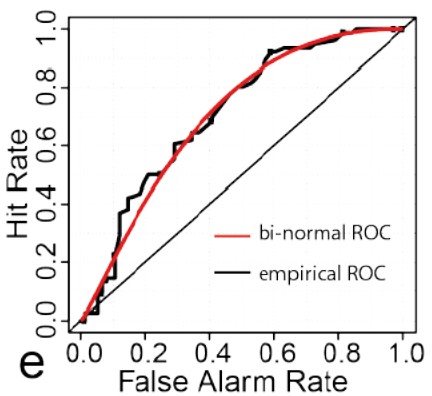

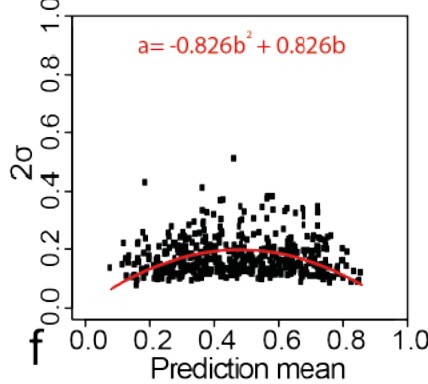

**Figure 4.** SU-based LR models prediction performance results: summary tables of the Cohen's kappa index, area under the ROC curve (AUC), overall accuracy ((TP+TN)/(TP+TN+FP+FN)) and overall error rate ((FP+FN)/(TP+TN+FP+FN)); (a,d) four fold or contingency plots; (b,e) ROC curves; (c,f) classification error plots and the quadratic regression fit curves (red line).

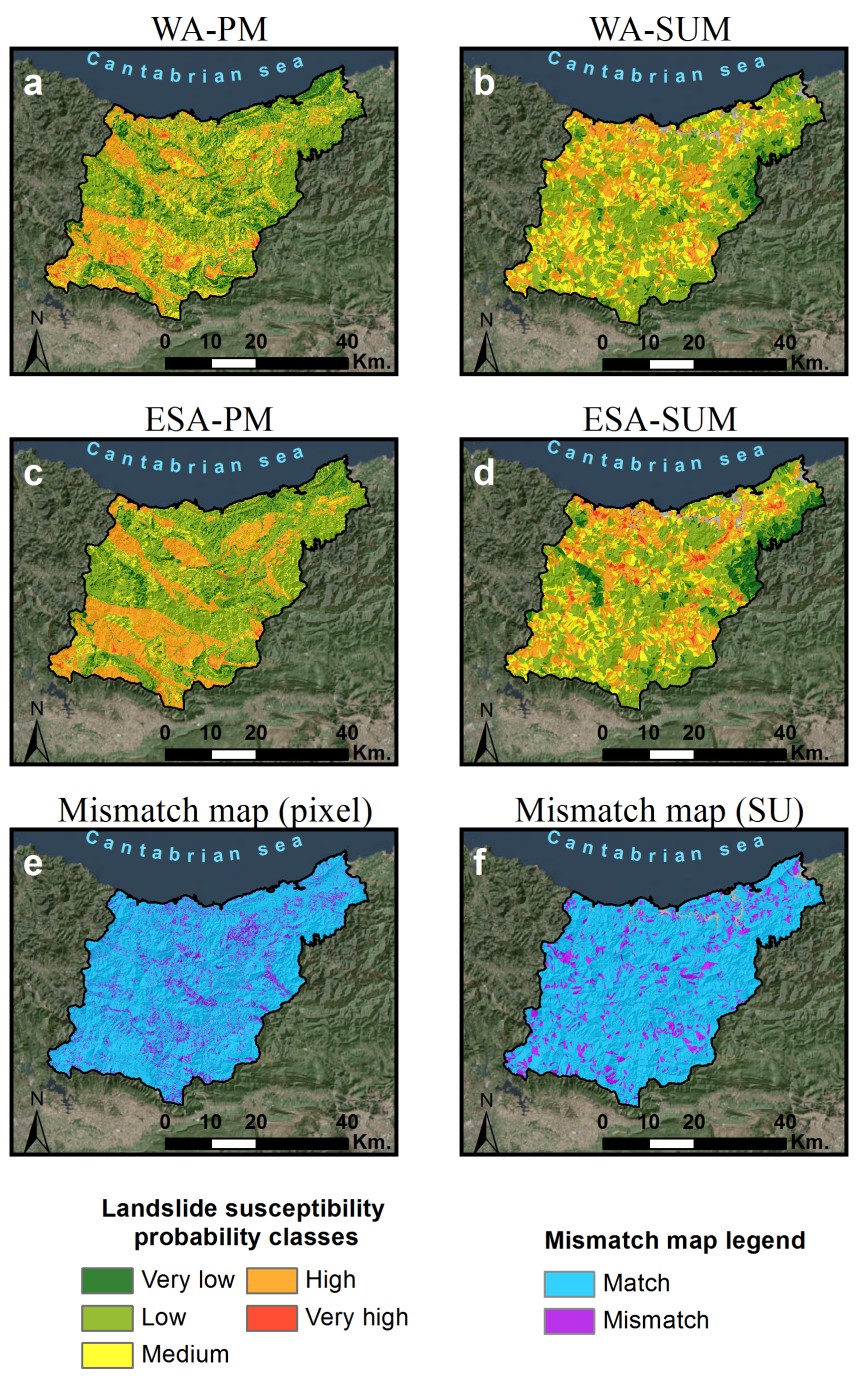

**Figure 5.** (a-d) Landslide susceptibility maps represented in five classes for WA-PM, WA-SUM, ESA-PM and ESA-SUM. (e,f) Mismatch maps representing the spatial distribution of the mapping units differently classified using ESA between pixel models and slope unit models.

**Table 1.** Results of the setting test of r.surbey in a 10 km$^2$ subset of the study area.

| Name | Resolution (m) | $D_{max}$ | Percentage of landslides within (%) |
|---|---|---|---|
| Survey 5 | 5 | 50 | 35 |
| Survey 6 | 20 | 50 | 70 |
| Survey 7 | 50 | 50 | 95 |
| Survey 8 | 100 | 50 | 100 |
| Survey 9 | 5 | 100 | 30 |
| Survey 10 | 20 | 100 | 60 |
| Survey 11 | 50 | 100 | 95 |
| Survey 12 | 100 | 100 | 100 |
| Survey 13 | 5 | 200 | 30 |
| Survey 14 | 20 | 200 | 55 |
| Survey 15 | 50 | 200 | 85 |
| Survey 16 | 100 | 200 | 100 |
| Survey 17 | 5 | 500 | 0 |
| Survey 18 | 20 | 500 | 35 |
| Survey 19 | 50 | 500 | 60 |
| Survey 20 | 100 | 500 | 95 |

**Table 2.** Set of environmental variables introduced for the whole area pixel-based (WA-PM) and effective surveyed area pixel-based (ESA-PM) models calculation, together with the significance p-value estimate corresponding to each explanatory variable (*cf.* Section 4.1). The best predictors were labelled with an asterisk.

| Name | Description | Significance p-value | |
|---|---|---|---|
| *Continuous* | | WA-PM | ESA-PM |
| Slope | The slope gradient in degrees. | $1.17 \cdot 10^{-189}$ | $1.06 \cdot 10^{-111}$ |
| Sinusoidal Slope | The sinusoidal mathematical transformation applied to the slope variable (Amorim, 2012) | $1.00 \cdot 10^{-155}$ | $7.57 \cdot 10^{-134}$ * |
| Surface area ratio | The relation between the theoretical volume and the surface of each pixel. | $3.743 \cdot 10^{-203}$ * | $1.89 \cdot 10^{-99}$ |
| Terrain wetness index | The spatial distribution of soil moisture or saturation (Yilmaz, 2009) | $9.864 \cdot 10^{-10}$ * | 0.126807342 |
| Curvature | The spatial variation of the slope gradient. | 0.909592654 | 0.525989188 |
| Plan curvature | The curvature of the surface perpendicular to the direction of the maximum slope. | 0.9094261 | 0.525836679 |
| Profile curvature | The curvature of the surface in the direction of the maximum slope. | 0.909605174 | 0.526032985 |
| *Categorical* | | | |
| Litology | The original categories have been reclassified by expert criteria (Geoeuskadi). | 0 * | 0 * |
| Permeability | The original categories have been reclassified by expert criteria (Geoeuskadi). | $1.496 \cdot 10^{-33}$ * | $7.632 \cdot 10^{-72}$ * |
| Regolith thickness | The layer for the study area has been obtained from the Lithological Map (Geoeuskadi). | 0 * | 0 * |
| Land Use | The original categories have been reclassified by expert criteria (Geoeuskadi). | $5.14 \cdot 10^{-291}$ | $1.42 \cdot 10^{-87}$ |
| Vegetation | The original categories have been reclassified by expert criteria (Geoeuskadi). | 0 * | $1.596 \cdot 10^{-173}$ * |
| Aspect | It represents the downslope direction measured in degrees classified in 9 classes. | 0 * | 0 * |