# Peer review of "Effective surveyed area and its role in statistical landslide susceptibility assessments"

_Natural Hazards and Earth System Sciences, 2018_

## Referee Comment (RC1) · M. Tonini (Referee) · 14 May 2018

This paper propose a procedure to delineate and to evaluate the performance of the effective surveyed area (ESA) in preparing a landslide inventory by geomorphological field mapping. A GRASS GIS python module was developed to define the ESA and the open source software provided as supplementary material. Landslide survey was performed in Gipuzkoa Province (Basque Country). Four computational domains were built-up: slope units as alterative to grid-cells for terrain subdivisions, and both the approaches were tested over the entire study area and over the effective surveyed area. Finally, landslide susceptibility maps and the associated uncertainties were assessed using multivariate logistic regression, while their classification performances were measured my means of a set of validation tests.

[Figure]

This paper focuses on an interesting topic, which is undoubtedly highly relevant in the domain uncertainty associated to landslide susceptibility mapping. It presents the new concept of effective surveyed area (ESA) and a new tool specifically developed to define this area. The overall presentation is well structured, the methods are appropriate, results are complete, accurate and reproducible.

However, the manuscript has some weak points and consequently it needs to be improved. Some sentences are too long and needed to be reformulate. The verb tenses need to be checked: sometimes past and present tense are used in an appropriate way.

My recommendation is that the manuscript could be accepted for publication by the journal Natural Hazard and Earth System Sciences with revisions outlined below in the attached supplement.

Please also note the supplement to this comment:
https://www.nat-hazards-earth-syst-sci-discuss.net/nhess-2018-88/nhess-2018-88-RC1-supplement.pdf

**Supplement:**

This paper propose a procedure to delineate and to evaluate the performance of the effective surveyed area (ESA) in preparing a landslide inventory by geomorphological field mapping. A GRASS GIS python module was developed to define the ESA and the open source software provided as supplementary material. Landslide survey was performed in Gipuzkoa Province (Basque Country). Four computational domains were built-up: slope units as alterative to grid-cells for terrain subdivisions, and both the approaches were tested over the entire study area and over the effective surveyed area. Finally, landslide susceptibility maps and the associated uncertainties were assessed using multivariate logistic regression, while their classification performances were measured my means of a set of validation tests.

This paper focuses on an interesting topic, which is undoubtedly highly relevant in the domain uncertainty associated to landslide susceptibility mapping. It presents the new concept of effective surveyed area (ESA) and a new tool specifically developed to define this area. The overall presentation is well structured, the methods are appropriate, results are complete, accurate and reproducible.

However, the manuscript has some weak points and consequently it needs to be improved. Some sentences are too long and needed to be reformulate. The verb tenses need to be checked: sometimes past and present tense are used in an appropriate way.

My recommendation is that the manuscript could be accepted for publication by the journal Natural Hazard and Earth System Sciences with some revision outlined below.

Page 1, line 18-21: move "Landslide susceptibility is defined as…" first and then introduce LSZ and provide its definition.

Page 2, line 2: specify that "spatial thematic layers" concern the predisposing factors.

Page 2, line 14: the sentence "image interpretation/classification expertise" it's not clear. You can modify for example with "accuracy classification performance due to uncertain factors".

Page 2, line 25: add some reference: which existing stdies?

Page 2, line 33-35: reformulate this sentence, which define the main goal of your study.

Page 3, line 2: I suggest to use "module" instead of "software", here as in the rest of the manuscript. But it's your choice.

Page 3, line 4 : "(i.e., the visible ….survey)" is redundant, as EAS  was already been defined before.

Page 3, line 5-10 : Reformulate this sentence. Some suggestions: "combining …with" (not "and"); define SU (slope units); you can provide information about the software that you used (LAND-SE) and the end: the fundamental think is that you used a "multivariate Linear Regression".

Page 3, line 29: which information about landslides location did you collect? Did you digitalized the entire perimeter? Or location refers to the four GPS points? You can explain this better.

Page 3, line 31: I would just say "most important to define the ESA,…."

Page 3, line 1: "information were then" (not "will be later")

Page 3, line 3: "using a combination": you can say better

Page 3, line 5: why did you consider only shallow landslides?

Page 4, line 12-17: Move the description of morphometric variable first, to respect the same order given in Table 1

Page 4, line 30: Change "defined" with "delimited"

Page 4, line 31: instead of "the software make use of" you could say "input data to define the visible area (i.e. ESA in our case) are the routes…."

Page 5, line 4: I would say "From this test" instead of "From the experiment"

Page 5, line 23-25: express all the areal values as square Km

Page 5, line 29: change with "by means of a multivariate logistic regression. Classification performances were …."

Page 5-6, line 31-4: Reformulate and simplify. Finally, you calibrated also the third and fourth map within ESA and WA and here is not clear at all.

Page 6, line 6: here it's not important that LR is implemented in LAND-SE software. Moreover, you already mentioned that you used this software to perform the analysis. What is important is that you applied the multivariate LR. Is it the most used statistical method for susceptibility in general of for landslide susceptibility? Please, specify.

Page 6, line 29: attention, you have never defined before the LR method that you are applying as a "multivariate binary classifier" and it's difficult for a reader to make the connection.

Page 7, line 12: explain how bootstrap technique works.

Page 7, line 20: I would say "In this study, the probability of landslide occurrence…."

Page 7, line 21: "was reclassified", instead of "can be"

Page 5-6, Section 4.2: You should introduce and discuss here the importance and need of defining a training and a validation set for statistical method, in a general way.

Page 7, line 31 and 34: "using pixel": change with, "grid calls (i.e. pixel based)", so it's more clear in the following of the text that you can use both these definitions.

Page 8, line 9. Why only three random set? What is the implication of using three or more?

Page 8, line 10-18: Reformulate and simplify, for example: "Finally, training and validation sets were selected as follow: 84,623 unstable pixels and an equal number of stable pixels for training; 29,247 unstable pixels and an equal number of stable pixels for validation. These two sets were selected at random first within WA and then within ESA."

Page 8, line 21: Are you sure you mean >0.15% of unstable pixels?

Page 8, line 25: move up to the same paragraph.

Page 8, line 25: do you mean that from your computation it results 304 unstable SUs? If so, please specify.

Page 8, line 26-30: Reformulate and simplify

Pagg 9, line 26 : explain what TWI is (i.e. Terrain wetness index); the same for SAR (line 29)

Page 9, line 32 and Page 10 line 4: why 152 validation MU? They are not 76?

Page 10, line 27: I would specify "multivariate linear regression"

Page 11: The obtained susceptibility maps (lines 5-9 and 19-24, Fig.5) could be introduced in the section 5. "Results"

Pag.17, Fig.1: You can combine the two in only one image. If you prefer to keep the two, in Fig.b the city is indicate only by the symbol and you can remove it.

---

## Referee Comment (RC2) · Anonymous Referee #2 · 15 May 2018

Dear Authors, I carefully read your paper about the role of effective surveyed area in susceptibility mapping and I think it is very interesting. You pointed out the need of a good knowledge about both the presence and the absence of landslides in the study area and you developed a software to define the ESA, where landslide presence or absence should be well known once the survey has been carried out. You also compared the susceptibility maps defined using different map units (grid cells and slope units) for different areas. In general, the paper is well written and clear, it has been properly structured and almost all phases of the work have been described; however, it needs to be improved. There are many issues that need to be fixed, before the paper can be considered for publication. Most of the cited works are self-citations or refer to co-worker papers, whereas several relevant papers from other scientists

about susceptibility mapping techniques have been ignored. It is not clear the utility of using the r.survey code to define the ESA, instead of using a simple portable GPS to record the surveyed area; its utility should be better described. The results of some experiments have been presented as they are, without providing any explanation or demonstration of their correctness (see sections 3.3 and 3.4). Main concerns regard the susceptibility mapping: why do you used different stability thresholds for WA and ESA? How you defined the threshold values? You cannot compare these maps if you used different criteria to perform the analyses. You should do a comparison using the same threshold value for both of them. One of the main outcomes is that the use of slope units increases the quality of the results, but the performance differences between the different approaches are very low; I believe that suck small difference do not justify the whole work. If a costs/benefits analysis would be performed, it could result that the WA-PM approach would be the best one. For the afore-mentioned reasons I believe that the paper has to undergo a major revision, before it can be considered for publication. Further comments are reported in the attached file.

Please also note the supplement to this comment:
https://www.nat-hazards-earth-syst-sci-discuss.net/nhess-2018-88/nhess-2018-88-RC2-supplement.pdf
* * *
[Figure]

**Supplement:**

[revised manuscript text omitted]

---

## Author Comment (AC1) · 4 Jun 2018

**NHESSD Interactive comments**

**Response to Review 1**

We thank the Reviewer for her detailed review of our work. We take note of your suggestions to grammatically improve the body text. However, in your specific notes there are some questions and comments that demand a direct response:

**Page 3, line 29: which information about landslides location did you collect? Did you digitalize the entire perimeter? Or location refers to the four GPS points? You can explain this better**.

We used the four GPS points collected on the field as reference to draw landslide polygons using QGIS software and Google Earth satellite imagery.

To clarify this point, we modified the text as follows:

"We collected information about the location of each observed landslide, four GPS points (crown, toe, and two flanks), photographs, surrounding area features and information about the landslide type, according to the Varnes, (1996) classification. Each documented landslide was drawn and digitized using its four GPS waypoints recorded and photographs as a reference. QGIS and Google Earth satellite imagery were used for the purpose."

**Page 3, line 5: why did you consider only shallow landslides?**

We considered that different typologies of slope instabilities are triggered by different mechanisms, which means that the predisposing factors and the way they affect to the occurrence of a given type of landslide can be different. Before the field campaign, we reviewed bibliographical sources, finding that within our study area 753 landslides were inventoried in different studies (INGEMISA, 1995; Gipuzkoako Foru Aldundia, 2013; IDE de Euskadi, 2014). Among all of them, 75% were considered as shallow slide type of movement, 10% were rock falls or rock mass deposits and 3% were flows or complex movements, adding to the 12% of the subset that was labelled as landslide scarp but without specifying the type of landslide. Although those bibliographical sources were not considered appropriate for our study – because they were heterogeneous in type and quality – they showed the most frequent type of landslide, so we focused our research on shallow landslides.

We included the information in the text as follows (section 3.1, page 4):

"During several field trips, 793 individual landslides were collected, and 746 of them were classified as shallow movements. Our observations together with the revised bibliographical sources (INGEMISA, 1995; Gipuzkoako Foru Aldundia, 2013; IDE de Euskadi, 2014) confirm that shallow slides are the most frequent type of landslide in the study area. Consequently, in order to consider only landslides triggered by the same mechanisms, only shallow movements were used as landslide presence when defining the dependent variable in the susceptibility assessment."

Gipuzkoako Foru Aldundia (2013). Evaluación y gestión integada de riesgos geotécnicos en la red de carreteras de la Diputación Foral de Gipuzkoa. Technical report, Mugikortasun eta Bide Azpiegituren saila, Unpublished report.

IDE de Euskadi (2014). Infraestructura de datos espaciales de euskadi.

INGEMISA (1995). Inventario y Análisis de las Áreas sometidas a Riesgo de Inestabilidades del Terreno de la C.A.P.V. Technical report, Eusko Jaurlaritza.

**Page 6, line 6: here it's not important that LR is implemented in LAND-SE software. Moreover, you already mentioned that you used this software to perform the analysis. What is important is that you applied the multivariate LR. Is it the most used statistical method for susceptibility in general or for landslide susceptibility? Please, specify.**

It is actually relevant to mention that we used LR as implemented in the LAND-SE software, since the latter is actually a comprehensive package for data preparation, model training and validation, and visualization of the results. Concerning LR, what the review paper discuss is only its application to landslide susceptibility studies. In the text, we modified "susceptibility" to "landslide susceptibility", to avoid ambiguity.

**Page 8, line 9: Why only three random set? What is the implication of using three or more?**

The test the Reviewer refers to was carried out just to confirm that the random selection of the landslide inventory would not affect the model results in a relevant way. Indeed, before starting with the main analysis, three preliminary LR runs were performed only changing the training and validation data sets. In all the cases the model classification performances were very similar. So, in order to choose only one data set for further comparative analysis, we decided to select the one with the best classification result, although we believe that conclusions would not be affected if any other data set would have been used.

We included the information in the text as follows (section 4.3, page 8):

"This exercise allowed us to confirm that the random selection of the landslide inventory would not affect the model results in a relevant way, because in all the cases the model classification performances were very similar."

**Page 8, line 21: Are you sure you mean >0.15% of unstable pixels?**

The total inventoried landslides actually covered 0.15% of the surface of the whole study area. Nevertheless, the presence of one single landslide pixel within a slope unit was not considered enough to label this SU as unstable. Therefore, instead of arbitrarily defining a given threshold value in order to consider a SU as unstable, we decided to use the overall landslide density in the WA.

We included the information in the text as follows (section 4.3, page 9):

"The presence of one single landslide pixel within a slope unit was not considered enough to label this SU as unstable. Therefore, instead of arbitrarily defining a given threshold value in order to consider a SU as unstable, we decided to use the overall landslide density in the WA. For this reason, we considered as unstable those SUs containing equal or more 0.15 % of unstable pixels, and stable otherwise."

**Page 8, line 25: do you mean that from your computation it results 304 unstable SUs? If so, please specify.**

Yes, in 304 cases the SU contained 0.15% or more unstable pixels.

**Page 9, line 32 and Page 10 line 4: why 152 validation SU? They are not 76?**

As it was explained in section 4.3 (Page 8, lines 25-30), 76 SU labelled as unstable were used for validation. Then, the validation sample was obtained by selecting the same number of SU labelled as stable. Thus, the validation sample contained 152 SU (76 unstable + 76 stable).

We included the information in the text as follows (section 4.3, page 9):

"In 304 cases the SU contained 0.15% or more unstable pixels, so we selected at random 228 of them (75%) for training, and the remaining 76 (25%) were used for validation. Like in grid cell approaches, we created two different training samples where unstable SUs were exactly the same, and only the stable SUs were different in each case. The first training sample includes 228 stable SUs selected at random along the WA. The second training sample includes an equal number of stable SUs units selected at random among those that at least partially overlap the ESA."

"Additionally, 76 SUs labelled as unstable were used for validation. Then, the validation sample was completed by adding a random selection of the same number of SUs labelled as stable and which at least partially overlap the ESA. Thus, the validation sample contained 152 SUs (76 unstable + 76 stable)."

---

## Author Comment (AC2) · 4 Jun 2018

**Response to Review 2**

We appreciate the Reviewer's comments about our research and we are pleased to see that our Manuscript was carefully reviewed. In the following we discuss the Reviewer's comments, including the suggestions on the bibliography and about adding new figures and tables. As a general remark, we would like to stress two points: (i) most of the references suggested by the Reviewer were known to us, and they were not explicitly included in the bibliography just because they appear in the revision papers cited, which represent a useful tool from this point of view; (ii) we believe that some comments contain misunderstandings about the very definition and meaning of the proposed idea of effective surveyed area (ESA), and occasionally they demand deeper explanations from our side. Thus, we will provide detailed answers to the Reviewer's comments below, and amend the Manuscript where needed.

**Answers to comments**

**It is not clear the utility of using the r.survey code to define the ESA, instead of using a simple portable GPS to record the surveyed area.**

A portable GPS device cannot be used to delineate areas, but only tracks or waypoints. During our data collection field campaigns we used GPS to record the route we followed. The point, here, is that even if one visits a given basin searching for landslide evidence, it is hardly possible to ensure that every single site of this basin was actually observed. This is due to the fact that some places can be not visible from the route followed during survey, typically existing roads. Thus, touring a basin does not ensure that the whole area was actually surveyed. This is the simple idea behind the ESA. The r.survey module is a tool that delineates the theoretically visible area from the points of view recorded during the field campaign by the GPS tracks. And, most importantly, the ESA, as delineated by r.survey, is an objective and reproducible portion of the study area directly observed by the geomorphologists, thus allowing to avoid arbitrary assumptions about which sites were actually surveyed and which ones were not.

**Why do you used different stability thresholds for WA and ESA? How you defined the threshold values? You cannot compare these maps if you used different criteria to perform the analyses.**

We did not use different stability thresholds. We carried out a few of the comparative analyses with a different stability threshold as an additional test, but we always performed pairwise comparisons between susceptibility maps obtained with the same threshold value.

The rationale behind testing two different thresholds is the following. The total inventoried landslides covered 0.15% of the surface of the whole study area (WA). Nevertheless, the presence of one single landslide pixel within a slope unit was not considered enough to label this SU as unstable. Therefore, instead of arbitrarily defining a given threshold value in order to consider a SU as unstable, we decided to use the overall landslide density in the WA for all the maps based on SUs, in order to consistently compare maps obtained training the susceptibility model within the WA and within the ESA. Additionally, as an additional control test, we prepared two additional susceptibility maps using as a threshold the landslide density within the ESA (0.33%). We observed that, even if the absolute value of the evaluation indexes (e.g., the $AUC_{ROC}$) slightly decrease with respect to the maps obtained using the threshold estimated in the WA (0.15%), the conclusions one can draw from the results of the two approaches are indistinguishable.

**One of the main outcomes is that the use of slope units increases the quality of the results, but the performance differences between the different approaches are very low; I believe that such small difference do not justify the whole work.**

In our opinion, pixel-based models and SU-based models are not simply comparable using absolute values of their validation performances, since they involve two different concepts of terrain subdivisions, with different meaning and application purposes.

The actual main question answered by our work is about the effect of training a statistical susceptibility model either in the portion of territory which was explicitly surveyed during a field campaign, the ESA, or in the extended area encompassing the ESA, the WA. As statistical methods are widely applied both using grid pixels and slope units, we addressed our question in both cases, since investigating only one of the two cases would not answer the question in the remaining one. Based on the results we obtained, we stated that pixel-based models (WA-PM an ESA-PM) showed more differences than SU-based models (WA-SUM and ESA-SUM). Our work is justified by the fact that we answered the question about relevance of the ESA, never addressed before in the literature, and not that one should use slope units instead of pixels as mapping units.

The small difference exhibited by the WA-SUM and ESA-SUM maps suggests that the use of SU as terrain partition mitigates the uncertainties introduced if the WA is used to calibrate the model. From this point of view, and from this point of view only, we conclude that SUs are best suited as a terrain partition approach. Of course the use of SUs has several additional advantages, but these are beyond the aim of the present work.

**If a costs/benefits analysis would be performed, it could result that the WA-PM approach would be the best one.**

To our knowledge, no similar analysis exists in the literature concerning landslide susceptibility studies. As a matter of fact, the very definition of costs and benefits associated with preparing landslide susceptibility maps is no easy task. Do we talk about the time spent on a single map, or we include the time during which a researcher builds his expertise, through his particular learning curve? Do we account for numerical models running time, and do we distinguish which software was adopted for the study? Do we talk about the economic cost of the equipment, and do we include the salary of the researchers? Do we distinguish between students and senior researchers? Not to talk about quantifying the benefits, that range from selfishly disseminating one's own scientific production, to actually providing the scientific community with useful tools, and even to save lives, since we are dealing with hazardous and potentially life-threatening natural hazards. In our opinion, one should stick to the fact that scientific advance is a step-wise process, and every single bit is helpful in producing a more effective and reliable research methodology that, hopefully, can be adopted by practitioners and decision-makers for natural hazard mitigation.

Concerning the specific object of our investigation, the difference in training a statistical model within the ESA or within the WA amounts to collect a few hundred GPS points during the field campaign, which represent a negligible time with respect to the time necessary to conduct the whole field survey, and run the r.survey software for a few seconds to obtain the ESA. Training the statistical model within the ESA does not produce overhead with respect to doing that within the WA. In conclusion, there only are advantages in using the ESA, since the "costs" are negligible.

Moreover, working with field based landslide data, the use of the WA as a calibration area is conceptually incorrect, because there may be places in which the presence or absence of

landslides must simply be inferred, with no actual evidence. By means of the ESA, we propose an alternative to revise this conceptual model, and our results suggest that this new approach produce performance advantages in validation.

**Answers to comments in the supplementary material**

**Page 1, line 22: "Please consider also some other works, as: Ermini et al. 2009 (10.1016/j.geomorph.2004.09.025), Catani et al. 2013 (10.5194/nhess-13-2815-2013), Yalcin 2008 (10.1016/j.catena.2007.01.003), etc"**

In page 1, line 22 we cited a review paper co-authored by one of us, in which "*an extensive database of 565 peer-review articles from 1983 to 2016*" was compiled. The large number of papers contained in the database is proof that the topic is widely discussed in the literature, and we believe that citing the review paper is the best thing to do, since citing a few papers about landslide susceptibility zonation would be a misrepresentation of the literature. Nevertheless, among the papers suggested by the Reviewer, we acknowledge that Ermini et al. 2009 (10.1016/j.geomorph.2004.09.025) is general enough to be included in the manuscript, also in consideration of its large citation records.

**Page 2 line 12: "Please considere also some other works, as Ardizzone et al. 2012,(DOI:10.1080/17445647.2012.694271), Rosi et al. 2018 (doi: 0.1007/s10346-017-0861-4), etc."**

As in the previous case, we believe that the references proposed by the Reviewer are not suitable in this specific part of the Manuscript, since they deal with two specific study areas and were aimed at producing specific landslide inventories, instead of investigating specific methodologies. Nevertheless, since the Reviewer points out that the list of references is not exhaustive of the literature, and consistently with what we stated in the previous comment, we prefer citing review papers in this specific point, namely:

Guzzetti et al., (2012) – already listed in the bibliography

Casagli N, Frodella W, Morelli S, Tofani V, Ciampalini A, Intrieri E, Raspini F, Rossi G, Tanteri L, Lu P. 2017. Spaceborne, UAV and ground-based remote sensing techniques for landslide mapping, monitoring and early warning. Geoenviron Disasters. 4(1):9. https://doi.org/10.1186/s40677-017-0073-1

And methodological papers, namely:

Fiorucci F, Cardinali M, Carlà R, Rossi M, Mondini A, Santurri L, Ardizzone F, Guzzetti F. 2011. Seasonal landslide mapping and estimation of landslide mobilization rates using aerial and satellite images. Geomorphology. 129(1–2):59–70. https://doi.org/10.1016/j.geomorph.2011.01.013

Fiorucci et al., (2018) – already listed in the bibligraphy

F Catani, P Farina, S Moretti, G Nico, T Strozzi. On the application of SAR interferometry to geomorphological studies: estimation of landform attributes and mass movements. Geomorphology 66 (1-4), 119-131. https://doi.org/10.1016/j.geomorph.2004.08.012

**Page 2, line 22: "Please consider also other papers. eg. Ba et al. 2018 (10.1007/s12145-018-0335-9), Zezere et al, 2017 (10.1016/j.scitotenv.2017.02.188)"**

We have added the references to the bibliography.

**Page 3, line 1: "It looks useless, since the surveyed area should be already know, once the survey is accomplished."**

This comment is somehow difficult to understand, since it refers to the (simple) code developed in this work to define the ESA, which is the central part of the Manuscript. We believe we have addressed this point at length before, in this response to the Reviewer, and demonstrated the usefulness of both the software, to delineate the ESA, and the effectiveness of the ESA, in calibrating statistical susceptibility models.

**Page 3, line 1: "This paragraph contains too many acronyms, it is hard to read. Please rephrase."**

We modified "*instead of the WA, enhances*" to "*instead of the WA (the whole study area, encompassing the ESA), enhances*" and removed LR, writing explicitly "*logistic regression*", so that the WA is explained right away and at least one acronym is removed.

**Page 3, line 23: "this is a generic sentence and the paper is about a case of study (central Italy). This citation is wrong and has to be changed. More relevant paper can be found in literature. Rainfall is the primary triggering factor of shallow landslides, not of rock fall, DSGSD, etc. "**

The citation is not wrong, since the whole cited paper deals with the effects of rainfall, and their projected changes, on landslide hazard for (rainfall-induced) shallow landslides. It is correct the cited study is about a case study in Central Italy, but it still is to the best of our knowledge the widest application of physically based slope stability models, in terms of study area extent. We also added one reference to a global analysis, Petley et al., (2005). Moreover, the Referee is more than right that rainfall is *not* the primary triggering factor for other types of landslides, such as rock falls, so we have modified "the primary triggering factor of landslides" to "the primary triggering factor of shallow landslides"

DN Petley, SA Dunning, NJ Rosser, O Hungr - Landslide risk management. The analysis of global landslide risk through the creation of a database of worldwide landslide fatalities Balkema, Amsterdam, 2005.

**Page 4, line 20: "the list of variable should be provided before this sentence."**

We have added the list of variables, in the revised version of the Manuscript.

**Page 5, line 4: "These result have to be proved in some way. I suggest that you add some graphs and table to support these results."**

We agree with the Reviewer that the description of the software and its usage should have been done in more detail. We added the following description and data in the Manuscript:

"In a 10 km$^2$ subset of the study area, we tested the software output using: i) maximum distance between sampled points of 50, 100, 200 and 500 m; ii) the original DEM at 5 m resolution and resampled versions of the DEM at 20, 50 and 100 m resolution; and maximum visible distance of 500 m (the later was dictated by the largest distance between the digitized field path and the farthest landslide pixel in the study area). Results of the test are summarized in Table 1.

| Name | Resolution | $D_{max}$ | Percentage of landslides within (%) |
|---|---|---|---|
| Survey_5 | 5 | 50 | 35 |
| Survey_6 | 20 | 50 | 70 |
| Survey_7 | 50 | 50 | 95 |
| Survey_8 | 100 | 50 | 100 |
| Survey_9 | 5 | 100 | 30 |
| Survey_10 | 20 | 100 | 60 |
| Survey_11 | 50 | 100 | 95 |
| Survey_12 | 100 | 100 | 100 |
| Survey_13 | 5 | 200 | 30 |
| Survey_14 | 20 | 200 | 55 |
| Survey_15 | 50 | 200 | 85 |
| Survey_16 | 100 | 200 | 100 |
| Survey_17 | 5 | 500 | 0 |
| Survey_18 | 20 | 500 | 35 |
| Survey_19 | 50 | 500 | 60 |
| Survey_20 | 100 | 500 | 95 |

As target criteria, we considered that the best setting option was the one which allows covering the totality of the landslides but using the less possible points (bigger $D_{max}$ value) and the lower possible resolution in order to optimize the calculation time.

In the case of the complete study area, the maximum visible distance was set in 1,100 m, in view that the largest distance between the digitized field path and the farthest landslide pixel was 1,092 m, and according to the results of Table 1, the rest of the settings were fixed: maximum sampling distance of 200 m, DEM resolution of 100 m."

Moreover, thanks to the revision process, we detected a mistake in the text of the manuscript. In page 5, line 5, it was written 100m as maximum sampling distance for the application of the r.survey code in the entire study area. The correct setting was of 2,000 meters, instead, as explained above. The mistake will be amended in the revised version of the Manuscript.

**Page 5, line 10: "Even if the equation is correct, I hardly can believe that a 2.6 sq.km landslide can be notice from 1 km distance. Maybe in arid or desertic areas, not in this area. Please clarify which is the maximum distance between landslides and ESA borders."**

We based our justification in the equation proposed by Rodrigues et al. (2010), that as a published scientific article we consider it reliable. Nevertheless, the smallest shallow slide in our inventory covers 7.3 $m^2$, which is more than the double area of the theoretical minimum area of an object to be visible from 1,100 m of distance (always according to Rodrigues et al. 2010). So, we considered the maximum visible distance of 1,100 m in r.survey settings was appropriate for our purpose. Moreover, we stated in the Manuscript that the equation serves to "make sense" of the orders of magnitude of the involved quantities. Obviously the simple equation represents an estimate, and neglects vegetation, geometric effects, and so on.

**Page 5, line 16: "I suggest that you remove from the code all the unnecessary comments, as unused strings, or comments in italian (if these comments are useful, please translate them).**
**Data preparation should be better descibed in the readme file (e.g. file format for points (shp, dxf, dwg, etc.) and DTM)."**

We fully agree on all the suggestions of the Reviewer and provided a clean version of the python code, along with a detailed description of its usage in the readme file. We apologize for the poor content of these pieces of work.

**Page 5, line 22: "These results (*of SU settings*) have to be supported by a more accurate description and, again, some graphs, tables, etc."**

We do not consider SU delineation as a "result" of the Manuscript, since the algorithm, software and optimization method were all described in already published pieces of work. To obtain an automatic delineation of the optimal SU partition in the study area, we used the the r.slopeunit software of Alvioli et al., (2016). We followed the optimization procedure illustrated at length in Alvioli et al., (2016), Schloegel et al., (2018) and Alvioli et al., (2018), so we did not include details in this Manuscript. For the sake of completeness, we can show in this Response to

[Figure]

Reviewers some of the details. Optimization of SU involves finding optimal values for the most relevant input parameters of r.slopeunits, namely "circular variance", $c$, and "minimum area", $a$. The search for optimal parameters and as a function of ($a$, $c$) is shown in the figure on the left. The figure shows an "aspect segmentation" metric $F(a, c)$, and the two bullets highlight the truly best (a, c) combination (blue) and the one selected for this study (green). The reason why we did not use the "optimal" (a, c) value is that we were working with a limited number of inventoried landslides,

and the large number if (smaller, on average) SU contained in the "optimal" subdivision would have led to a critical unbalance between landslides SU and no-landslides SU. So, we selected an SU subdivision with analysis smaller number of polygons, labelled "Large SU" in the Figure.
We do not believe this information actually would add to the content of the paper; we actually believe that it would be off-topic and make the Manuscript's main message difficult to follow. We consider SU delineation as a technical step and we would not like to include this discussion in the Manuscript.

**Page 6, line 7: "If it is the most used, maybe more than 1 work can be cited, you cited only 1 review paper."**

We have here the same motivations as in the previous occurrence of this citations: see comments above. Moreover, in this particular point, we explicitly quote one of the results of the review paper, i.e. that logistic regression is the most used method in the literature, so we are confident that this is the correct citation. Moreover, it is not true that it is the only work we cited: in fact, one line below, we mentioned that logistic regression "proved to be useful" in several studies, among which the three further works we have cited (Nefesliogluet al., 2008; Van Den Eeckhaut et al., 2012; Trigila et al., 2015).

**Page 6, line 16: "How do you accounted categorical variables in this equation?"**

We addressed this issue in section 3.2 when we state that: "*For categorical variables, we computed frequency ratio (FR) values for each class, and used them as a relative value for their transformation into continuous variables (Lee and Min, 2001; Yilmaz, 2009; Trigila et al., 2015)*".

So at the end, we accounted for categorical variables in the LR model's equation in the same way as continuous variables.

**Page 6, line 23: "Did you perform a Student t-test? Please clarify how the p-value is calculated."**

We apply statistical models contained in the LAND-SE software of Rossi and Reichenbach (2016). The software is an R script, and the logistic regression results are obtained using calls to $glm$ (generalized linear model) function. The implementation of the $glm$ function in R is such that it is possible to investigate the estimated standard error of a t-statistic for the null hypothesis of each of the coefficients of the linear model. The p-value represents the probability that the parameter is zero: for p-values much smaller than 0.05 the null hypothesis (vanishing coefficient) is rejected, thus the associated variable is significant for the final result.

**Page 7, line 1: "Please add more references"**

We have the same comment as above. Moreover, the mentioned metrics are very well-known in the community and need not be supported by additional references, in our opinion.

**Page 7, line 15: "Why did you use 2 sigma, instead of 1? Please clarify"**

We used 2σ following Guzzetti et al., (2006). The test represents an estimate of the variations obtained when input data is changed at random among many different, equally possible, data sets. Accounting for a 2 σ variation in a Gaussian model means accounting for 95% of the area under the probability density function. In this case, it represents the unit of variations in each pixel, or for each slope unit, which makes rather arbitrary the choice of 2 σ, σ or anything in the same order of magnitude.

**Page 8, line 9: Why use three random sets of training/validation partition?**

The test the Reviewer refers to was carried out just to confirm that the random selection of the landslide inventory would not affect the model results in a relevant way. Indeed, before performing the main analysis, three preliminary LR runs were performed only changing the training and validation data sets. In all the cases the model classification performances were very similar. So, in order to choose only one data set for further comparative analysis, we decided to select the one with the best classification result, although the preliminary test shows that conclusions would not be affected if any other data set would have been used.

**Page 8, line 21: 0.15%? Are you sure? This value is very, very low. Please clarify how you defined this value.**

See the second paragraph in Answer to comments section.

**Page 8, line 25: Stable and unstable SUs have the same numbers (228 Unstable and 228 stable SUs for training, 76 for validation). Please check these values.**

The choice of an equal number of stable and unstable SUs was done on purpose, and it is the standard procedure required by the LAND-SE software for landslide susceptibility assessment. The logistic regression model requires a balanced dataset, in which the number of stable and unstable cases are similar (Costanzo et al., 2014). There are other studies, like Felicísimo et al., (2013), that already tested the influence of introducing a larger number of negative data (no-landslide locations). They concluded that this strategy decreases the performance of the models, which is intuitively understandable. For this reason, in order to complete the training

and validation samples, the same amount of SU defined as stable (and randomly selected) was added to each subset. So the training sample contained 456 SUs (228 unstable + 228 stable); and validation sample was composed by 152 SUs (76 unstable + 76 stable). We adopted the same procedure for the pixel based susceptibility maps. We have added the following reference to the bibliography:

Felicísimo, Á. M., Cuartero, A., Remondo, J., & Quirós, E. (2013). Mapping landslide susceptibility with logistic regression, multiple adaptive regression splines, classification and regression trees, and maximum entropy methods: a comparative study. Landslides, 10(2), 175-189.

**Page 8, line 32: I believe you should use only those units that are totally within the ESA, you do not know what is going on outside the ESA.**

We already considered this issue, because we agree that this way would be conceptually more consistent. However, there are some crucial aspects that have to be taken into account on this respect:

- The ESA is an automatic approximation of the real surveyed area that is likely to be contained in the actual ESA. Thus the fact that a given portion of a SU stay outside the ESA does not necessarily mean that it was not observed.
- Excluding all the SU that strictly were not within the ESA rejected most of the SUs in the training sample, because that prescription would include the SUs exceeding the ESA by a little portion. The opposite situation in which most of the SUs fall outside the ESA in our case occurred fewer times than the other way around.
- In those SUs where landslides were actually observed, they should be automatically considered for the model even if a portion stay outside the ESA, because the presence of instabilities was already confirmed, so the presence or not of instabilities in such theoretically not observed space would change nothing.

Thus, for these reasons, we concluded that the approximations of either excluding or not those SUs that falls partially within the ESA would produce an effect smaller than the difference between the results obtained by training the statistical model within the ESA or within the WA.

We tried to explain this idea in page 8, lines 33 - 35 when we state that "*if a portion of an SU falls within the ESA, that implies that at least one part of the SU was observed. Therefore, using this approach, we can remove at least those SUs that were not surveyed at all*".

**Page 9, line 8: From table 1, it results that you did not use topographic parameters (slope, curvature, etc.) in WA-PM. For ESA-PM the only used continuous parameter is senoidal slope. Is it correct? How can you explain it? I suggest to perform some comparisons with susceptibility maps obtained with different sets of variable (all variables, same set of WA map, etc.) to verify if you selected the best set.**

In table 1 we showed with an asterisk the final explanatory variables introduced in WA-PM, and two of these are the Surface Area Ratio and the Topographic Wetness Index, which are both topographic and continuous variables. In ESA-PM, the only continuous variable selected as final predictor was the Senoidal Slope. Results of the correlation tests were not included in the manuscript, but they indicated that Slope, Senoidal Slope and SAR are highly correlated between each other, so according to the procedure explained in section 3.2, only the one with the lowest p-value was selected as final predictor. Following the same rationale, TWI was rejected in ESA-PM because of its high p-value.

We applied this simplified and statistically oriented work-flow in order to objectively define the final set of predictors to be introduced in each pixel-based model. We maintain that our final aim is to show the effectiveness of training a statistical model within the ESA as compared to

training within the WA, and not to obtain the "perfect" input data set or the "best" susceptibility map, as we stated in the Conclusions. We argue that the comparison is meaningful as long as it is performed between maps obtained consistently with the same input data sets and conditions, pairwise. As a matter of fact, the number of ways the two susceptibility maps in a pairwise comparison can be prepared is virtually infinite. Nevertheless, as we mentioned in the Abstract and in the Conclusions, we strongly believe that our results apply to any similar map pairs obtained with meaningful input data sets and statistical methods, as long as the significance of the statistical analysis is ascertained.

**Page 9, line 22: This is not an objective approach. I suggest to perform some comparisons with susceptibility maps obtained with different sets of variable (all variables, same set of WA map, etc.) to verify if you selected the best set.**

Se the next answer below.

**Page 9, line 29: If Senoidal slope is derivative of slope, why do you not use slope in WA analysis? It should have the same significance.**

According to the observations carried out by Santacana et al. (2003) and Amorim (2012), there are some types of landslides, like shallow slides, typically occurring in medium slope areas, decreasing their presence from 45º of slope on. Such a behaviour can be explained with the lack of surface formations in very steep areas, since rocks outcrop in such areas. Thus, depending on the type of landslides considered to the susceptibility analysis, the relation between them and the slope may not be completely linear and positive, because in some cases, from 45º of slope, the more is the slope the less is the probability of finding landslides. Considering that our landslide inventory is about shallow slides, we decided to consider the senoidal transformation of the slope as possible explanatory variable for landslide susceptibility modelling, according to Santacana et al. (2003).

Following the objective procedure to select explanatory variables explained in section 3.2, in ESA-PM senoidal slope was selected over slope because its p-value vas lower. In WA-PM instead, SAR was selected as most suitable explanatory variable over slope and senoidal slope for the same reason.

In the particular case of the SU-based models, explanatory variables are organized in a different way due to the irregular size of the terrain units. We used the mean and the standard deviation of the morphometric variables within each SU as explanatory variables, and the percentage of the area covered by each class of the categorical layers. This made impractical the application of the same variables selection approach, and we acknowledge that a specific variable selection approach for SU models would require further investigations. Nevertheless, in order to ensure the use of significant and non-redundant variables we decided to select at least those variables that were always selected in the pixel-based models. Then we added Slope as a morphometric variable because we considered it more suitable to describe the average morphology within a SU than the Senoidal slope or the SAR.

Santacana, N.; De Paz, A.; Baeza, C.; Corominas, J. y Marturià, J. (2003) A GIS-based multivariate statistical analysis for shallow landslide susceptibility mapping in La Pobla de Lillet area (Eastern Pyrenees, Spain). Natural Hazards 30(3):281–295.

**Page 11, line 19: You should clarify how you defined the susceptibility classes reported in fig. 5.**

This explanation actually is in page 20, lines 20-23. "*The probability of landslide occurrence resulting from each model estimate (trained either within the ESA or within the WA) and for each considered mapping unit (either grid cells or slope units), can be reclassified in five landslide susceptibility classes which were labelled as Very low (for susceptibility values in the range 0-0.2), Low (0.2 0.45), Medium (0.45-0.55), High (0.55-0.8) and Very high (0.8-1)*".

**Page 11, line 31: How can you state that the landslide inventory you used in not complete? How is it possible? This point is crucial, please clarify**

We state in page 12 line 24 that the moderate prediction capacity of the resulting susceptibility maps could be, among other reasons due to **"***the lack of more complete landslide inventory*", but it doesn't mean that our landslide inventory was considered incomplete. Even though we acknowledge that more landslides than those included in our inventory probably exist, we maintain that the data collected by direct field observation offers very reliable information, and considering that the landslides size distribution is in agreement with what is expected from a complete inventory (Figs. 2b and c; see Malamud et al. (2004)). We believe that our landslide inventory is complete for the purpose of this research that, we stress, is about showing the effectiveness of training a statistical model within the ESA as compared to training within the WA, and is not about obtaining the "perfect" input data set or the "best" susceptibility map. We added the following text and references the revised Manuscript:

*Completeness refers to the proportion of landslides shown in the inventory compared to the real (and most of the times unknown) number of landslides in the study area* (Guzzetti et al. 2012; Malamud et al., 2004).

BD Malamud, DL Turcotte, F Guzzetti, P Reichenbach (2004). Landslide inventories and their statistical properties. Earth Surface Processes and Landforms 29 (6), 687-711 https://doi.org/10.1002/esp.1064

**Page 12, line 8: This is not trure: in fig.4, 3 out of 4 parameters of WA-SUM are better than those of ESA-SUM.**
**-Does training ares correspond to ESA? If they are different you should provide a map of the training area.**

We rephrased this paragraph as follows:

"A straightforward comparative analysis using standard prediction performance metrics revealed that the ESA-based approach is better than the WA-based, at least in grid-cell mapping units based approaches, for what concerned the training area (i.e. within the ESA or WA). Introducing different mapping units in the comparison, we further found that using slope units slightly reduces the gap between results obtained training a statistical model within the ESA versus WA. Thus, the capacity of the slope unit mapping subdivision to mitigate this error was demonstrated, as suitable alternative to the conventional pixel-based approaches."